# Enablers and Barriers of Accessing Health Care Services among Older Adults in South-East Asia: A Scoping Review

**DOI:** 10.3390/ijerph19127351

**Published:** 2022-06-15

**Authors:** Nurul Syuhada Mohd Rosnu, Devinder Kaur Ajit Singh, Arimi Fitri Mat Ludin, Wan Syafira Ishak, Mohd Harimi Abd Rahman, Suzana Shahar

**Affiliations:** 1Centre for Healthy Ageing and Wellness (H-Care), Faculty of Health Sciences, Universiti Kebangsaan Malaysia, Kuala Lumpur 50300, Malaysia; p111597@siswa.ukm.edu.my (N.S.M.R.); arimifitri@ukm.edu.my (A.F.M.L.); wsyafira@ukm.edu.my (W.S.I.); suzana.shahar@ukm.edu.my (S.S.); 2Optometry and Vision Sciences Programme, Center for Rehabilitation and Special Needs, Faculty of Health Sciences, Universiti Kebangsaan Malaysia, Kuala Lumpur 50300, Malaysia; harimirahman@ukm.edu.my

**Keywords:** accessibility, enablers, barriers, healthcare, older adults, South-East Asia

## Abstract

South-East Asia (SEA) is the home of the largest number of the world’s older population. In this scoping review, we aimed to map the existing enablers and barriers of accessing healthcare services among older adults in SEA countries. Articles that were published from January 2001 until November 2021 were searched in four data sources (PubMed, Web of Science, EBSCO Host and The Cochrane Library). Studies pertaining to the factors which assist or obstruct older Southeast Asian adults from assessing healthcare services were chosen for this scoping review. First, two reviewers screened the titles and abstracts of articles in the data sources. After identifying appropriate articles, the reviewers read them. Data extracted by one reviewer were verified by the other reviewer. The findings were then classified according to Penchansky and Thomas’s five domains of access. A total of 19 studies were included in the final scoping review. Accessibility and acceptability were the two factors most often identified as enablers or barriers to older adults from accessing healthcare. Other often mentioned factors were finances, transportation and social/family support. Older adults living in rural areas were especially impacted by these factors. To promote healthy ageing, optimum healthcare and wellbeing among older adults in Southeast Asia, it is extremely important to consider accessibility and acceptability when planning healthcare services.

## 1. Introduction

In 2050, there will an estimated 1.5 billion persons aged 65 and over globally. In the Southeast Asian (SEA) countries, the proportion of people aged 60 years and above has been predicted to account for 13.7% and 20.3% of the population in 2030 and 2050, respectively [1]. East and Southeast Asia are home to 260 million of the world’s older population. The high proportion of older individuals in SEA means that the cost of providing them with healthcare services will be high [2]. This is significant as only two of the eleven (11) southeast Asian countries (Brunei and Singapore) have high income status. The other nine (Cambodia, Indonesia, Laos, Malaysia, Myanmar, Philippines, Vietnam, Timor-Leste and Thailand) have varied income statuses [3].

Ageing is commonly associated with progressive loss of skeletal muscle mass and decrease in metabolism function and functional capacity [4]. The other challenges include chronic illnesses, sensory impairments (vision and hearing), risk of falls and mobility and cognitive decline [5]. The disability burden among older adults globally is primarily driven by functional decline, vision and hearing loss, and pain [6]. Despite the physiological changes that lead to limitations, evidence have revealed that older adults can indeed age successfully [7]. However, it may be a challenge for any healthcare system to maintain the health and wellbeing of this vulnerable group. It is particularly challenging for most SEA countries, which have limited fiscal resources, infrastructure and qualified healthcare workers, to care for their ageing populations.

Access is an important component in health policy and has been defined as having “the timely use of personal health services to achieve the best health outcomes” [8]. Access to healthcare results from the interface among persons, households, social and physical environments and health care systems [9]. To date, there have been numerous frameworks used to conceptualize access to healthcare. The commonly used access frameworks are Andersens’ Behavioural Model of Health Services Use, Frenk’s framework, Levesque’s Conceptual Framework for Healthcare Access and Penchansky and Thomas’ framework. The access to healthcare according to Penchansky and Thomas’ framework portray access as the degree of “fit” between clients’ needs and the healthcare system [10].

Generally, in most SEA countries, the public pension system is considered weak, with inadequate benefits and coverage for old-age economic security [11]. As a result, increased individual, family and societal burden occur due to increased healthcare needs with ageing. In addition, healthcare utilization is affected by various factors such as poverty and accessibility [12]. Difficulties in accessing healthcare-related services result in unmet healthcare needs, delayed care and poor management of chronic illnesses, leading to increased emergency room visits [13].

Although the quality of the healthcare system is one of the key drivers to healthcare access, obtaining it with ease is equally important. In the Philippines, despite universal health coverage that has been actively pursued since 2010, disparity in the availability and accessibility of healthcare resources are still prominent within regions as they are mainly located at the urban cities [14]. Health expenditure in Indonesia is mainly (60%) out-of-pocket due to the low share (39%) of the Indonesian government’s health expenditure [15]. Such issues could represent barriers to getting healthcare services, especially among older persons. In this case, constant changes and fine-tuning to adapt to the sociodemographic and economic changes are required; Singapore’s healthcare system could be the best for benchmarking [16].

Access to healthcare is a global concern. For example, in Australia, the limited number of healthcare employees, lack of services and poor infrastructure have been identified as barriers to accessing healthcare services [5]. About 55% of Americans considered access barriers as one of the reasons for delayed or missed care [17]. Access barriers include difficulties obtaining appointments, poor transportation and limited office hours. It can be assumed that these situations are more common in SEA countries, as most of them have low to middle income statuses.

When planning holistic healthcare solutions, information regarding the barriers and enablers that impact older adults’ accessibility to healthcare services are important. Thus, this review explores the barriers and enablers that affect older adults in SEA countries accessing healthcare services.

## 2. Methods

This scoping review sought to answer the following research questions: “What factors enable older adults to access healthcare services in SEA countries?” and “What factors obstruct older adults from accessing healthcare services in SEA countries?” To be thorough and systematic, this scoping review adopted Arksey and O’Malley’s [18] framework, except for optional Step 6 (consultation with stakeholders). The Preferred Reporting Items for Systematic Reviews and Meta-Analyses Extension for Scoping Review (PRISMA-ScR) was used as a guideline.

### 2.1. Search Strategy

A literature search was conducted in November 2021. After the search strategy was developed, it was independently verified by two reviewers. The following electronic databases, PubMed, Web of Science, EBSCO Host and The Cochrane Library, were searched systematically to identify relevant studies published between January 2001 and November 2021. The identified studies were first screened for eligibility based on information contained in their titles and abstracts.

The search terms used for this review were:Older population OR Elderly OR Ageing population OR GeriatricEnabler* OR Enabling OR Factor* OR Facilitator OR Motivator*Barrier* OR Limitation* OR Restriction OR Challenges OR DifficultyHealth Service* OR Health care OR HealthcareSouth-East Asia OR Brunei OR Burma OR Myanmar OR Cambodia OR Timor-Leste OR Indonesia OR Laos OR Malaysia OR Philippines OR Singapore OR Thailand OR Vietnam1 AND 2 AND 3 AND 4 AND 5

### 2.2. Selection Criteria

All studies that reported issues with older adults accessing health services, regardless of study design (except for letters to the editor and conference proceedings with abstract only), were eligible for this review. This scoping review included only studies conducted among adults (aged 50 years and above). All types of health care services were included.

### 2.3. Selection of Included Publications

The search outcomes from each database were first screened for eligibility by reviewing the studies’ titles and abstracts. The studies were then read completely to further screen for suitability based on the review questions and inclusion criteria. Figure 1 outlines the selection process. Experts were not consulted for this scoping review.

### 2.4. Data Extraction and Analysis

NS and AF sorted the included papers according to Penchansky and Thomas’s [19] five dimensions of access: availability, accessibility, accommodation, affordability and acceptability, which are defined as follows:Availability compares the volume and type of available healthcare services with the demand for said services. Availability refers to whether healthcare providers and facilities (including clinics and hospitals), including providers of specialized services such as mental health and emergency care, can adequately supply the said services.Accessibility refers to the location of healthcare services and the location of the demand for said services, considering distance, travel cost, and time to travel to these services.Accommodation pertains to how healthcare services meet the demand for their services (for example, appointment systems, operation hours, walk-in facilities, telephone services).Affordability describes the cost of healthcare services and the availability of insurance and/or other means to pay for said services.Acceptability refers to whether the parties demanding healthcare services accept the personal and other characteristics of the healthcare providers (including age, sex, location, type of facility, religious affiliation) and trust the healthcare providers.

## 3. Results

### 3.1. Literature Search

Figure 1 shows a PRISMA flowchart for the selection of papers for this scoping review. Searches of the four databases yielded 2149 articles after duplicates were removed. After screening the titles and abstracts of the articles, it was determined that the majority (*n* = 2109) of these articles did not meet the inclusion criteria. Forty papers were identified as potentially relevant, but 21 papers did not pertain to the issue of accessibility to healthcare services in SEA countries and were discarded. A total of 19 articles were included in this review. Data regarding study design, age, country, type of healthcare service, main findings and the five dimensions of access were then combined into Table 1.

### 3.2. Characteristics of the Included Studies

All the publications (*n* = 19) used in the final data extraction were published between 2001 and 2021. The included studies were from Cambodia (*n* = 2), Indonesia (*n* = 3), Malaysia (*n* = 2), Philippines (*n* = 1), Singapore (*n* = 2), Thailand (*n* = 6), Timor-Leste (*n* = 1) and Vietnam (*n* = 2). Of the 11 South-East Asia countries, studies regarding enablers and barriers in accessing healthcare services for older adults were found in eight countries. Brunei and Burma were the exceptions.

Study design, age, country, type of healthcare service, and main findings of the studies that met the inclusion criteria are summarized in Table 1. Thirteen studies used quantitative methods, while four studies used qualitative methods of data collection. Only two studies used a mixed-methods approach involving interviews and focus group discussions, with data relevant to issues of accessibility to healthcare services presented quantitatively and qualitatively.

The results from the included papers were then categorized according to Penchansky and Thomas’s [19] five dimensions of access, as shown in Table 2. Of the 19 papers included in this review, 11 identified accessibility issues, ten identified acceptability issues, eight identified affordability issues and two identified availability and accommodation issues. Papers that had the respective dimensions of access discussion are marked ‘X’ 

#### 3.2.1. Availability

The issue of availability was discussed in two studies [20,36]. The studies indicated that availability of healthcare services in rural areas is still a significant issue. The disparity in the availability of mental health services in the rural areas of Vietnam was reported as a barrier [36]. The availability issue was further discussed in terms of the limited number of health care professionals in rural areas. An under-supply leading to the unavailability of doctors was also highlighted by older adults in the Philippines as one of the reasons for their health care needs not being met [20]. This may be due to the high numbers of skilled health professionals moving from rural to urban areas [31].

#### 3.2.2. Accessibility

Accessibility issues were reported in more than half the studies (58%) included in this review [20,21,22,25,26,29,30,33,35,36,38] The review determined that the accessibility issues faced by older adults in SEA countries primarily concerned geographic, financial and social/family support factors. In a Thai population, urban dwellers were more likely than dwellers in rural areas to use health care services [25]. The distance an older adult had to travel to access health care services was also an important determinant, as a major reason for unmet health care needs was that the health care services were too far away to access [20,24,32].

Rittirong [33] found that the likelihood of an older adult not using a health care service increased by about 30% with an increase of every kilometer they had to travel to said health care service. In contrast, Kullanit & Taneepanichskul [26] found that older adults who lived far away from healthcare services were more likely to use them. In addition, a strong correlation between transportation costs and the use of healthcare services was highlighted [26]. The higher the transportation costs, the less likely older people were to use healthcare services. This indicates that travel cost and travel distance are important factors older adults consider when deciding whether or not they should access healthcare services.

Several studies included in this review highlighted that, in terms of family and social support, older adults who are married and/or have children are more likely to use healthcare services [25,26]. One of the reasons for unmet health needs was “there is nobody to take me to hospital,” which is cited often by older adults living alone and older adults living with only one family member [29].

#### 3.2.3. Accommodation

Accommodation issues were reported in two studies [22,24]. The operating hours of government-run healthcare services were noted as a barrier to older adults accessing healthcare services. Older adults in Cambodia found the hours to access government-run dental services inconvenient [22].

#### 3.2.4. Affordability

The issue of affordability appeared in eight of the included studies [21,22,28,29,32,36,37,38]. Older adults whose incomes are below national poverty levels are more likely to have unmet healthcare needs, compared to those whose incomes are above national poverty levels [29]. The older adults’ inability to pay for healthcare services is a highly significant issue. In a recent Indonesian study that examined the association between economic barrier (insurance ownership) and access to health services revealed that 15% of very poor older persons have major access barriers to health services [38].

#### 3.2.5. Acceptability

Acceptability was discussed in majority of the included studies [23,24,26,27,28,29,30,31,34,35]. Older adults’ perceptions on their health condition and healthcare services may motivate or hinder them from accessing services. The older adults’ perceptions of their health condition and healthcare services motivate or hinder them from accessing healthcare services. This review found that the older adults’ perceptions, age, sex, type of facility and the healthcare providers’ attitudes played important roles in determining whether they accessed healthcare services.

Two studies found that trust in healthcare professionals is an important factor [24,27]. Older Cambodians expressed distrust in their local dentists as one of their reasons for not seeking dental care. The studies found no strong association between perceptions and satisfaction in hospital visits among older adults [26]. This may indicate that older adults perceive the maintenance of their general health as more crucial compared to the maintenance of their mental and dental care.

Based on the qualitative studies that were included in this review, the main themes that emerged as factors that may act as enablers or barriers to access health services were: (1) transportation, (2) financial, (3) family/social support, and (4) perceived need for care. Similar to the quantitative findings included is this review, the themes transportation, financial and family/social support factors can be explained in the accessibility domain of Penchansky and Thomas [19]. In terms of perceived needs for care, older adults’ perception towards dental and eye health were lower compared to general health. Fear, anxiety and past negative experience with dentistry care were the main factors appearing as barriers to access dental clinic. For instance, an older adult shared “I am always worried because I am very scared of tooth polishing which I had the last time.” [30]. In addition, a low perceived need to visit dental clinic due to the absence of teeth, which is prevalent among older adults, was stated as “Now I have no teeth, so no need to visit a dentist.” [30]. The theme ‘perceived need for care’ that emerged in the qualitative studies [30,31] could be explained based on the acceptability domain.

## 4. Discussion

This scoping review aimed to map the literature concerning factors which enable and obstruct older adults in SEA countries from accessing health care services. To the best of our knowledge, this is the first such review concerning access to healthcare among older adults in SEA countries. In the past ten years there has been increasing interest in healthcare access for older adults as countries prepare for aging populations. Our findings will assist policymakers and stakeholders as they develop plans to cater to the needs of older adults, which would include improving their access to healthcare.

The present review presented the issues of access to healthcare according to Penchansky and Thomas [19]. This scoping review suggests that access to healthcare services for older people is a construct between individuals and services. Service accessibility, availability and individual acceptability play important roles in facilitating or hindering older people’s access to healthcare services. The availability and accessibility of healthcare services may not be sufficient to encourage older people to access them. Moreover, older people should not be treated as a homogeneous group [39], as factors that act as enablers or barriers for a particular individual may not apply to other older persons.

The studies included in the scoping review indicate that accessibility and acceptability are the two accessibility domains most often discussed. Financial, geographical and social/family support are viewed as interactive factors that determine older adults’ accessibility to healthcare services. Older people with no formal employment tend to have only limited pension or no pension at all. Throughout SEA, there are numerous innovative, pro-poor financing schemes to increase coverage of basic health services for the vulnerable and disadvantaged groups. For instance, Thailand has its Health Card and 30-baht Schemes, Vietnam has the Health Fund for the Poor, Cambodia and Laos have their Health Equity Funds, and Singapore has its Medifund [40]. In Malaysia, older adults can assess public health services with little cost. Moreover, the nominal fee is subsidized for government pensioners and employees [41]. However, older adults with lower socioeconomic status may still be unable to afford these services [42]. Older adults in Indonesia are covered by the National Health Insurance and, in 2015, approximately 54.58% of Indonesian older adults were covered by this insurance scheme [43].

This review identified that, despite these health financing schemes, older adults in SEA countries still face financial barriers when accessing healthcare services. In Indonesia, older adults not covered by insurance have decreased access to healthcare services, leading to unmet healthcare needs and negative health outcomes [28]. The “wealth-health” gradient, which highlights the positive relationship between health and wealth, becomes more pronounced as people age [44]. Since financial resources are directly proportional to healthcare access and thus health status, more efforts are required to support older adults and over-burdened healthcare systems.

Societal and family support also play important roles in enabling older people to access healthcare. Several of the included studies that reported older persons’ unmet health needs focused on older persons living alone or living with only one family member. In Asia, there is a long-standing custom of parent-grandparent co-parenting [45]. A few studies reported the positive effects on older people’s psychological and well-being when they are cared for by their children [46]. However, as the older people would be taking care of their grandchildren in exchange, they may be hindered from going for health checks [36].

This review identified older persons’ perceptions, age, sex, type of facility, as well as the healthcare provider’s attitude as important factors in increasing access to healthcare services. Older people’s perceptions are important in determining acceptability. This is especially evident in mental health services, which relate to stigma and other emotional concerns. In Vietnam, mental illness is defined as severe psychiatric abnormality or madness that can also afflict others [36]. Negative perceptions of mental health services increase the difficulties for older people to access mental health services.

Figure 2 illustrates the framework of factors that may act as enablers or barriers for older adults’ access to healthcare adapted from the Aday and Andersen (1974) model [47]. The framework describes the interaction among five domains: (1) health policy, (2) characteristics of the health delivery system, (3) characteristics of the individual, (4) health services utilization, and (5) consumer satisfaction. This review found that health policy, such as financing, manpower and organization, are the key drivers in ensuring access to healthcare among older adults. Health policy has a unidirectional effect on health delivery system and individual characteristics. The health delivery system characteristics, which are levels of healthcare (preventive or acute/rehabilitation) and types of healthcare (medical, dental, and mental health), have an influence on the utilization or access health services.

Utilization or access to health services was the main interest of this review. Within the Aday and Andersen (1974) [47] model, we found that individual characteristics and consumer satisfaction towards services are the main factors that may act as enablers or barriers to access among older adults in this review. This finding is similar to an international report, where the access and utilization to health services were greatly influenced by health delivery system features, consumer satisfaction and individual characteristics [48].

The findings from this review have important implications for both healthcare service providers and healthcare policymakers. SEA is a region with more than half a billion people, diverse cultures and a rapidly ageing population. In order to tackle future challenges related to an ageing population, knowledge of older adults’ accessibility to health care services should be addressed to better provide healthcare for this vulnerable group. For instance, identifying specific issues older persons encounter when accessing healthcare services can facilitate planning and allocating the right resources, for example, identifying locations for the provision of future healthcare services, revising operating hours, and making health insurance more affordable.

With the recent COVID-19 outbreak, older adults’ access to healthcare was highly jeopardized. Limited contact with friends, family and caregivers disrupted the older adults’ daily routines and access to healthcare [49]. As a result, telerehabilitation became an alternative platform to deliver health education and care [50]. Despite the majority of older adults being digitally illiterate, a recent review study found that older adults found telerehabilitation acceptable, thus improving access to healthcare [51,52]. However, the older adults’ acceptability and barriers in implementing telerehabilitation should be addressed first to improve the achievement of targeted outcomes. Providing preventive healthcare services closer to home or basing them in the community would make them more accessible for older adults [52]. Such services could be used to empower and engage older persons for self or home care [26,50]. 

This review maps out the issues faced by older adults in SEA countries when accessing healthcare services. Although the search was comprehensive, there are some limitations. Only 19 studies that were relevant to this review objectives were included. While this may be insufficient to conclude a reliable conclusion, it is an indicator that more research is needed in this area for evidence to strategize and meet the health needs of older adults. In addition, grey literature, which may provide additional insights, was not searched. This review was also limited to citizens of SEA countries. Migrants/refugees were excluded from this review. Thus, given the inadequate health care access by migrants or refugees in SEA countries, it is recommended that further studies investigate healthcare accessibility of migrants/refugees groups.

## 5. Conclusions

This review summarized the issues that act as enablers or barriers for older adults to access healthcare services in SEA countries. Numerous issues were identified as enablers or barriers to access health services, according to the individuals’ socioeconomic, living status and geographical area. Understanding the current healthcare access for older persons is crucial for policy makers and providers to formulate and implement policies and services that are applicable to Asian cultures, life experiences and social circumstances.

## Figures and Tables

**Figure 1 ijerph-19-07351-f001:**
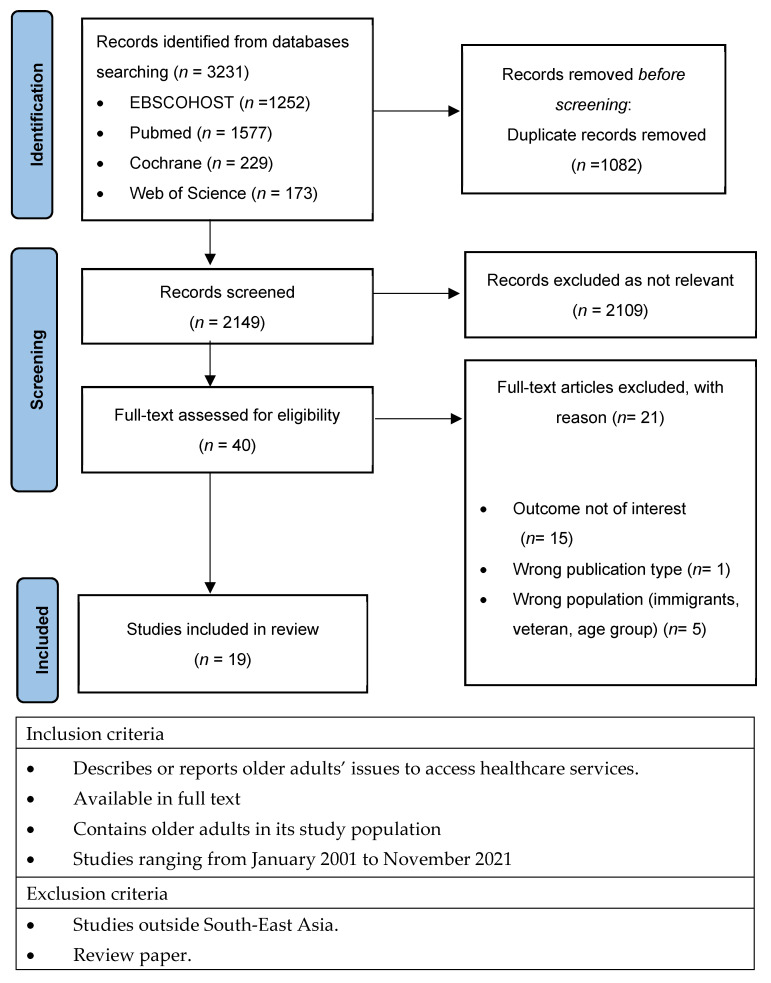
PRISMA flow diagram. The PRISMA diagram details the search and selection process applied during our systematic literature search for this scoping review. PRISMA, Preferred Reporting Items for Systematic Reviews and Meta-Analyses.

**Figure 2 ijerph-19-07351-f002:**
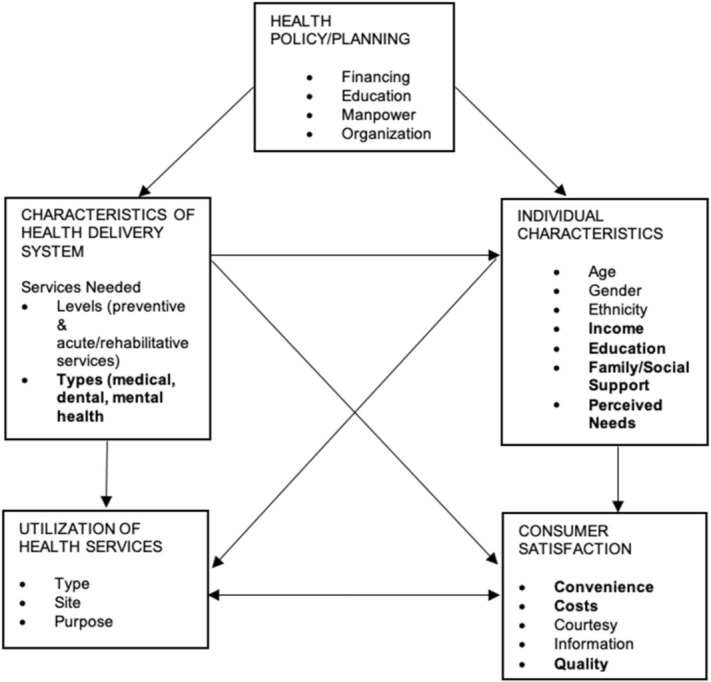
Conceptual framework of access adapted from Aday and Andersen (1974) [47]. Factors highlighted in bold are the main findings from this review.

**Table 1 ijerph-19-07351-t001:** Summary of included studies.

Author/Year/Country	Study Design	Objective	Population Age	Service Type	Outcome Types	Findings
Carandang et al. (2019) [20] Philippines	Qualitative study	To examine the perceptions of unmet needs and to explore the coping mechanisms of senior citizens	60–85 years	Health care services	Unmet need for health care services	Themes: Staffing problem, drug supply problem and accessibility—health centre is far away from home
Goh (2011) [21] Singapore	Mixed method study	Understanding the family care of elderly person and their use of post-acute care services	65 years and over	Post-acute care services (Community hospital, Nursing home, Day Rehabilitation Centre)	Predisposing factor Enabling factor Need factor	Individual variables such as ethnicity, family size, paid help and housing type meditate use and draw attention to underlying financial barriers.
Horn R et al. (2018) [22] Cambodia	Qualitative study	To explore the oral health experiences, practices and perceptions of older adults in Cambodia.	60 years and over	Oral health	Social Environment Physical Environment Use of services	Physical environment themes: Problems with transport to health facilities. Use of Service themes: Cost as a main barrier, inconvenient operating hours
Irwan et al. (2016) [23] Indonesia	Quantitative study	To examine self-care practices and health-seeking behaviours of older adults in urban Indonesia	60 years and over	Health care services	Predisposing factor Enabling factor Need factor	Respondents with higher self-efficacy, those who did not want to get information, and those of younger ages are less likely to visit health centre regularly
Kamsan et al. (2021) [24] Malaysia	Quantitative study	To determine the healthcare utilization and its associated factors among older persons with knee OA	60 years and over	Outpatient, inpatients and pharmacotherapy	Predisposing factor Enabling factor Need factor	Being married and having an income is associated with the usage of outpatient services and pharmacotherapy
Kanthawee et al. (2014) [25] Thailand	Mixed method study	To describe perception towards health and social services among elderly people in Chiang Rain province	60 years and over	Health care services	Barriers to access on health services	Cost of transportation and unviability of family members. Difficult to assess hospital because hospital is quite far from their home
Kullanit and Taneepanichskul (2017) [26] Thailand	Quantitative study	To examine the association between healthcare utilization and transportation barriers and perception among elderly in Thailand	60 years and over	Health care services (health promoting hospital, district hospital and provincial hospital)	Transportation barrier (travel duration, distance from home, have company, expenses) Accessibility of healthcare facilities on transportation	Travel duration and distance from elderly home to healthcare services was associated with healthcare utilization. Elderly satisfaction and perception on ability to pay for transportation expense was related to their healthcare utilization.
Lee et al. (2013) [27] Timor-Leste	Quantitative study	To identify and compare the barriers in using eye care services	Adults aged 40 years and above	Eye Care Services	Barrier to use eye care services	Participants aged ≥60 years were more likely to claim they were too old, there was no-one to accompany them or fear as barrier in using eye care services
Madyaningrum et al. (2018) [28] Indonesia	Quantitative study	To identify factors related to the use of outpatient services among the Indonesian elderly	60 years and over	Outpatient services	Predisposing factor Enabling factor Need factor	Economic status, health insurance, self-rated health, region and number of chronic conditions were associated with use of outpatient services.
Meemon and Paek (2020) [29] Thailand	Quantitative study	To explore incidence rate of and reasons for unmet health needs for older adults living alone	65 years and over	Health care services	Socio-economic characteristics on unmet health needs	For older adults living alone and living with one family member, lack of caretakers and quality of care was one of the major reasons for unmet health needs. Those living with more than one family members reported quality of care in the hospital as reason for unmet health needs.
Mittal et al. (2019) [30] Singapore	Qualitative study	To understand factors affecting dental care utilisation among older Singaporeans who are eligible for CHAS or PG subsidies	65 years and over	Dental care	Barriers to visit dental clinic	Fear, anxiety and past negative experience with dentist, do not perceive need to visit dentist and lack of awareness
Neyhouser et al. (2018) [31] Cambodia	Qualitative study	(i) To identify consumer and provider barriers for women accessing eye health care. (ii) To identify provider barres for wo	45–84 years (mean age 63)	Eye care	Perceptions regarding access barriers to eye health care	Gender-based differences in decision-making, access and control over resources and women’s social status contribute to impeding women’s access to eye health services
Quashie and Pothisiri (2018) [32] Thailand	Quantitative study	To explore the rural-urban gaps in health care utilization among older Thais.	50 years and over	Health care services	Predisposing factor Enabling factor Need factor	Predisposing factors and health needs narrow the rural-urban gap while enabling factors widened the gap in health visits.
Rittirong J (2019) [33] Thailand	Quantitative study	Factors predicting health centre visits among elderly with chronic illness	60 years and over	Health care services	Living arrangement Geographic and modes of travel	Elderly living with adult child more likely to visit health care.
Samsudin and Abdullah (2017) [34] Malaysia	Quantitative study	To identify the determinant of health care utilisation by the elderly in northern states of Malaysia	60 years and over	Doctor visit and inpatient stays	Predisposing factor Enabling factor Need factor	Only age and use of alternative healthcare have roles in determining likelihood of doctor’s visit. Inpatient determinants; (i) Socio-economic factors (ii) Gender
Thammatacharee et al. (2012) [35] Thailand	Quantitative study	Assess annual prevalence, characteristics and reasons for unmet healthcare need in the Thai population	60 years and over	Outpatient and inpatient	Socio-economic characteristics	The most prevalent reasons for unmet outpatient healthcare needs among those aged 60 years and above are (i) too far to travel, (ii) not sure these are effective treatments and (iii) ‘other reasons’. Inpatient unmet reasons: (i) No accompanies, (ii) ‘other reasons’
Van et al. (2021) [36] Vietnam	Quantitative study	Prevalence of associated factors among the elderly living in rural Vietnam	60 years and over	Mental health care services	Socioeconomic information Perceived social support Perceived Barriers to Psychological Treatment (PBPT)	Foremost barriers to psychological service are (i) emotional concerns about psychological services, (ii) geographic and financial difficulties, (iii) participation restrictions
Nguyen and Giang [37] (2021) Vietnam	Quantitative study	To examine factors affecting the use of healthcare services using Anderson’s Behavioural Model	60 years and over	Public and private healthcare facilities	Predisposing factor Enabling factor Need factor	Having to pay medical costs was the major reason for older people not using all kinds of healthcare facilities
Laksono et al. (2018) [38] Indonesia	Quantitative study	To examine the barriers for the elderly in accessing health services	50 years and above	Health care services	Travel time, transportation cost, insurance ownership	Elderly living both in urban and rural have moderate access barrier to health center. 15% of very poor elderly have major access barrier.

**Table 2 ijerph-19-07351-t002:** Healthcare access issues of included studies according to Penchansky and Thomas’ five dimensions of accessibility.

Publication	Availability	Accessibility	Accommodation	Affordability	Acceptability
Carandang et al. (2019) [20]	X	X			
Goh (2012) [21]		X		X	
Horn R et al. (2017) [22]		X	X	X	
Irwan et al. (2016) [23]					X
Kamsan et al. (2021) [24]			X		X
Kanthawee et al. (2014) [25]		X			
Kullanit at Taneepanichskul (2017) [26]		X			X
Lee et al. (2013) [27]					X
Madyaningrum et al. (2014) [28]				X	X
Meemon and Paek (2020) [29]		X		X	X
Mittal et al. (2019) [30]		X			X
Neyhouser et al. (2018) [31]					X
Nguyen and Giang (2021) [32]				X	
Quashie (2018) [33]				X	
Rittirong J (2019) [34]		X			
Samsudin and Abdullah (2017) [35]					X
Thammatacharee et al. (2012) [36]		X			X
Van et al. (2021) [37]	X	X		X	
Laksono et al. (2018) [38]		X		X

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
