# Peer review of "Enablers and Barriers of Accessing Health Care Services among Older Adults in South-East Asia: A Scoping Review"

_ijerph, 2022, doi:10.3390/ijerph19127351_

Round 1

Reviewer 1 Report

Thank you for the opportunity to review this paper.  The authors have presented an interesting paper with potentially useful results for future planning within health area services in South-East Asia. Below are some comments intended to strengthen the manuscript:

  • In your introduction as you set up the rationale for the scoping review you focus on illness and healthcare costs associated with again. Of course, this is relevant to your rationale, however there is a risk to only presenting the deficit model of aging.  Consider also noting the gains and strengths associated with aging to reduce risk of ageism and apocalyptic demography.
  • I wonder if it would be helpful to provide some context around the various healthcare systems in SEA?  You give examples of Australia and the US but access is also influenced by the model of the healthcare system (e.g. universal, etc).
  • Was an information scientist consulted in the development of the search strategy? Is the more information re: MeSH terms or keywords?
  • Given the region of focus I’m curious about the rationale for only including English articles. Perhaps this is something that can be unpacked in the limitations if there are many articles that were missed that might have been relevant?
  • Since you rely so heavily on Penchansky and Thomas’s (1981) five dimensions of access I think this paper would be strong with some additional context and overview of these (and why this is useful tool for organizing your analysis) in the intro / rationale
  • Please note in Table 1 the way study designs are categorized is not consistent (e.g. qualitative, quantitative or mixed methods vs. cross-sectional
  • I think it’s important to note you used a deductive approach (I assume?). Did you also do any inductive analyses?  That is, did you create and identify any themes relevant to research questions that fall outside of the five dimensions of access?
  • The results section feels quite thin. Discussion is somewhat stronger but overall felt like there were opportunities for additional reporting and meaning making.

Reviewer 2 Report

The paper is well written and easy to read. Older adults’ access to health care services is of significant importance in aging society. Since numerous studies have looked into this issue, it is meaningful to conduct a review on how these studies find in their case studies. However, in its current status, the paper needs further improvements before possible publication.

1. Only 18 studies were included in the final review. This may be not enough to reach a reliable and meaningful conclusion from the scoping review. The authors need to make sure whether all relevant studies have been searched and included. Besides, they can include other studies not published in English, if any, in the review. Otherwise, they need to carefully explain why the sample of 18 studies can support a scoping review.

2. The review is focused on SEA countries. What are the specific characteristics, e.g.., institutional, economic, socio and urban planning-related factors, for SEA countries compared to other countries across the world? On the other hand, whether SEA countries share common characteristics to a certain extent? If not, why is it meaningful to treat them as a specific group of countries in terms of health care service system.

3. In the discussion, the authors tried to summarize the enablers and barriers of accessing health care services included in the reviewed studies. But only simple descriptions were given, and it is hard to understand how the findings are different from the international literature. Therefore, the contribution of the study is unclear. The authors are suggested to develop a thorough conceptual framework assembling the enablers and barriers factors revealed by the review. This framework is meaningful for the understanding of factors influencing health care services for older adults in the specific context of SEA countries. Moreover, the framework and corresponding factors should be compared to the international literature to highlight the characteristics in SEA countries.
